# Level of Depression during the COVID-19 Pandemic in Poland—A Cross-Sectional Study

**DOI:** 10.3390/healthcare10061123

**Published:** 2022-06-16

**Authors:** Anna Zalewska, Monika Gałczyk, Katarzyna Van Damme-Ostapowicz

**Affiliations:** 1Department of Physiotherapy, Faculty of Health Sciences, Lomza State University of Applied Sciences, Akademicka 14, 18-400 Lomza, Poland; aanna.zalewska@gmail.com (A.Z.); monikagalczyk@onet.eu (M.G.); 2Department of Health and Caring Sciences, Faculty of Health and Social Sciences, Western Norway University of Applied Sciences, Svanehaugvegen 1, 6812 Førde, Norway

**Keywords:** COVID-19, depression, Beck Depression Inventory, adults, Poland

## Abstract

Objectives: The aim of this study was to assess the level of depression during the COVID-19 pandemic in Poland. Methods: The online survey was conducted among Polish adult citizens (204 respondents) with a positive SARS-CoV-2 test result. The level of depression was assessed by the Beck Depression Inventory in Polish. Results: Depressive symptoms of moderate or severe degree appeared in about every fourth person. Women were characterised with higher mean depression scores. In the group of men, significant correlations were found between mental condition and age—the higher the age, the higher the values of depression measures. Those who were asymptomatic with SARS-CoV-2 had the best results—a lack of depression, while those who were fully symptomatic had the worst results—major depressive disorder. Conclusions: There is a need for further research and monitoring of mental health in specific population groups. It is necessary to plan preventive measures to prevent the negative effects of the pandemic, especially in women. Specialist support should be implemented during and after the pandemic.

## 1. Introduction

COVID-19 disease, designated as severe acute respiratory syndrome-2 (SARS-CoV-2), first appeared in December 2019. An outbreak of the plague was detected in Hubei province in Wuhan [1]. With more cases in neighbouring countries, the disease began to spread to other countries around the world and was declared a public health emergency of international concern by WHO in late December 2019 [2]. In Poland, the first confirmed case of coronavirus infection was detected in March 2020 [3].

The COVID-19 pandemic brought many changes to the lives of people all over the world. It forced them to adapt their daily activities and habits to completely new conditions. These changes were strongly influenced by restrictions on movement, isolation, as well as changes in the nature of on-site work and study to off-site. These phenomena were mainly aimed at combating the spread of the virus and its consequences—severe courses of chronic diseases and a high incidence of deaths [4,5,6,7,8]. 

Unfortunately, these measures have not always brought the desired effects. The virus is still present in our lives and attacks, at various times, to a greater or lesser extent, with long-term negative effects, not only functional but also emotional. It can contribute to changes in the human psyche, including the emergence of generalised fear, increased anxiety or even the occurrence of depressive syndromes requiring the use of professional psychological care [9,10]. Both the pandemic and the various activities aimed at its prevention, such as quarantine and social distancing, can have a negative impact on mental health. There are many people around us who struggle to cope with a new pandemic situation and, therefore, are exposed to higher levels of stress and anxiety [11]. 

A review of the literature shows that there are a number of causes that may influence the onset of anxiety or even depression [12,13,14]. During the COVID-19 pandemic, increased depressive symptoms became a natural reaction of the body to the uncertainty and overwhelming sense of threat that became the new everyday reality [8]. A pandemic has a negative impact on mental health as it can lead to stress-related and life-threatening illness, economic crisis, unemployment or fear of losing family members [3]. Research on mental health during pandemics is extremely important as this problem affects many people worldwide [15,16]. Documenting mental health disorders allows us to identify those people who are at risk, and to explore psychological and social resources that can reduce their risk [17]. For this reason, our study aimed to assess the level of depression in a sample of the Polish population. The authors focused on gender differences, age, course of SARS-CoV-2 infection (if any) and possible need for hospitalisation. 

Before 2019, there were approximately 83 million people living with depression in Europe, of which approximately 10% were Polish [18]. From estimates obtained before the pandemic, depressive conditions were mostly diagnosed in Poland [19]. 

In view of the pandemic situation and its health consequences, there is a need for research into the impact of the pandemic on mental health. Therefore, the aim of this study was to assess the level of depression during the COVID-19 pandemic in Poland. The implementation of the main objective was based on the search for answers to the following research questions, which are also specific objectives: Is there any association between the course of infection and the need for hospitalisation and depression? Is there a relationship between gender, age and depression?

## 2. Materials and Methods

### 2.1. Participants and Procedure

In August 2021, during the COVID-19 pandemic, a cross-sectional online survey was conducted among Polish adult citizens with a positive SARS-CoV-2 test result. The survey, in the form of a link to a Google form, the information about the study, as well as anonymity and voluntary consent to participate were sent via the researchers’ institution’s social media profile (Facebook) and the researchers’ university emails. Respondents followed the authors’ instructions when completing the questionnaire, in which the presence of at least 6 out of the 10 most common symptoms (fever, cough, myalgia, olfactory disturbances, dysgeusia, general weakness, fatigue, diarrhoea, difficulty in breathing or dyspnoea, headache) was considered as full-symptomatic COVID-19. Similarly, the presence of up to 3 of the 10 most common symptoms (other than difficulty breathing or dyspnoea) was considered oligosymptomatic COVID-19, without symptoms—asymptomatic. Respondents also answered whether they were tested for SARS-CoV-2 and whether the result was positive or negative. The authors found this method to be the most effective and fastest. From the received questionnaires, the subjects who did not meet the inclusion criteria for the study were discarded. Every fifth person was randomly selected. A random number generator with a range of 1–100 was used, and those individuals for whom the generated number was between 1 and 20 were included in the analysis. This way, questionnaires were obtained from 204 individuals (122 women and 82 men). The studied population is a representative sample for the population of recovered patients in Poland within the age range from less than 25 years to more than 55 years.

Inclusion criteria for the survey were: age over 18 years, consent to participate in the study, testing for SARS-CoV-2 and a positive test result for SARS-CoV-2. Exclusion criteria were: age under 18, lack of consent to participate in the study, failure to test for SARS-CoV-2 and a negative test result for SARS-CoV-2.

The research was approved by the Senate Commission for Ethics in Scientific Research of the School of Medical Science in Bialystok KB/162/2020/2021. The respondents voluntarily participated in the survey and the course of the research was published on the Personal Data Protection Act of 10 May 2018 (Journal of Laws of 2018, item 1000), in accordance with the Regulation of the European Parliament and the Council (European Union) 2016/679 as of 27 April 2016, on the protection of individuals with regard to personal data processing and on the free movement of such data, and the repeal of Directive 95/46/EC (General Data Protection Regulation). Respondents were informed about the study objectives, the interview process and the applicable data protection guidelines. 

### 2.2. Methods of Assessing Depression

The Beck Depression Inventory (BDI) was created by Aaron T. Beck [20]. It is a self-completion tool that consists of 21 questions. Over the years, it has become one of the most widely used questionnaires to measure the severity of depression. The BDI total score is an arithmetic summation of all 21 symptoms scored on a final scale from 0 to 63. The higher the total score, the higher the level of depression [21]. Depending on the number of points obtained, the results are also classified into a 4-point adjective scale:0–11—no depression;12–26—mild depression;27–49—moderate depression;50–63—severe depression [22].

A validated inventory in Poland was used for the study [23].

The value for Cronbach’s alpha reported in papers is >0.7 [24,25,26,27].

### 2.3. Statistical Methods

The relevance of differences among groups was assessed using the Kruskal–Wallis test—such statistics and this test were chosen because of the significant asymmetry of the measures considered. The correlation of age with the level of mental condition was examined using Spearman’s rank correlation coefficient. A significance level of *p* < 0.05 was established for all statistical analyses.

## 3. Results

### 3.1. General Characteristics

The survey was conducted on 204 people (Table 1). Women predominated among the respondents (Table 2).

### 3.2. Information on Contact with SARS-CoV-2 Virus

The largest proportion of respondents (48.5%) experienced COVID-19 asymptomatically. Only every fourth person assessed the symptoms during the disease as complete (Table 3).

More than 90% of respondents were not hospitalised because of COVID-19 transmission (Table 4).

### 3.3. Severity and Level of Depression (by Beck)

The Beck questionnaire was used to assess the level of depression in the studied population. The questionnaire was not completed by 17 respondents. The mean level of the measure of depression in the study group was about seven points, and the median was three points (Table 5).

Classification of the level of depression according to the suggested norms was performed (Table 6). The summary only applies to those who completed the questionnaire. It turned out that most respondents did not suffer from depression. Symptoms of moderate or severe depression appeared in about every fourth person.

### 3.4. Course of Infection and Depression

The key issue for health status should be the course of the disease. There are statistically significant differences in the Beck Depression Scale, with the best scores in people who were asymptomatic, worse mental condition in those with few symptoms, and the worst scores in those who were fully symptomatic (Table 7).

### 3.5. Need for Hospitalisation and Depression

In addition, those who were hospitalised showed worse depression than those who were not hospitalised for COVID-19 (Table 8). However, in part, this may be a result—as in the previous summary—of the fact that the more severe course of the disease occurred in relatively older people.

### 3.6. Gender and Level of Depression 

There is large variation between men and women (Table 9).

### 3.7. Age and Level of Depression

The analysis was conducted separately for the female and male groups. Statistically significant correlations of depression with age were found in the male group. The higher the age, the higher the values of Beck depression measures (Table 10). 

## 4. Discussion

Studies conducted over the past two years have shown that depressive symptoms are common during the COVID-19 pandemic [28,29,30]. In our study, we found that among the respondents, 23.0% suffered from moderate depression, while 2.7% from severe depression.

For the Beck Depression Scale, there are statistically significant differences, with the best scores in people who were asymptomatic or did not have SARS-CoV-2 infection, worse mental health observed in people with few symptoms, and worst scores in people who were fully symptomatic. In addition, those who were hospitalised showed a worse mental condition than those who were not hospitalised for COVID-19. However, no cause–effect relationships can be established due to the cross-sectional nature of the study.

A similar study, conducted online in Poland in 2020, using the Beck Inventory, found that women had higher mean depression scores [3]. This overlaps with our study, which observed that there was a large gender gap in depression, and that women were more likely to have higher mean scores. Another online study conducted among women in Poland in 2021 also confirms that respondents presented mental health difficulties in the form of anxiety and depression [31]. At this point, it is important to recall the WHO report from 2020, which alerted that women are particularly vulnerable to the negative impact of the COVID-19 pandemic on mental health [32]. Contemporary literature also confirms the need to study the mental condition of women, who are more likely to suffer the negative consequences of unpredictable events [33,34,35].

In our study, a statistically significant correlation of mental condition with age was found in the male group—this may be due to the greater age diversity in the respondents of this gender (there was a large group of young men). The older the man, the higher the values of Beck’s depression measures. Interestingly, results obtained by other researchers indicate that elderly women coped better with functioning in the new reality associated with the COVID-19 pandemic [36]. In a study by Idzik et al., elderly women in Poland also showed lower severity of depressive symptoms [31]. A study conducted in a Polish cohort of 1179 people indicates that people over the age of 64 years have statistically significantly lower levels of depression than those aged 18–24 and 25–35 years, and that the age of the subjects was negatively correlated with symptoms of depression severity [37].

The strengths of the presented survey were: easy access to the surveyed group and low cost of the conducted research. Unfortunately, the survey also had some limitations, which were: the small size of the surveyed group impeding the examination of the prevalence of depression at the country level and the subjectivity of answers. Respondents were also not asked a question about the official method of diagnosing SARS-CoV-2 virus; they only answered whether they had a test performed and whether it was positive or negative. It was also an online survey, which was conducted by providing a link to the survey, which limited access to the study for those without internet access. In addition, confounding factors influenced the study. They may not have consistently reflected the information that was obtained from the survey. Because this is an observational study, it cannot be concluded that depression is causally related to COVID-19. 

## 5. Conclusions

There is a need for further research and close monitoring of the psychosocial functioning of various social groups, especially in the event of a prolonged pandemic and the emergence of further waves of COVID-19. It is essential to plan preventive measures aimed at preventing the negative effects of the pandemic on mental health. This prevention should be especially targeted at women. There is also a need to implement psychiatric and psychological assistance during and after the pandemic, in order to mitigate its psychological consequences.

## Figures and Tables

**Table 1 healthcare-10-01123-t001:** Age of respondents.

Age	Number	Percentage
<26 years	50	24.5%
26–35 years	49	24.0%
36–45 years	49	24.0%
46–55 years	46	22.5%
56–65 years	7	3.4%
>65 years	3	1.5%

**Table 2 healthcare-10-01123-t002:** Division according to sex.

Sex	Number	Percentage
woman	122	59.8%
man	82	40.2%

**Table 3 healthcare-10-01123-t003:** Mode of infection transmission.

Mode of Infection Transmission	Number	Percentage
asymptomatic	54	26.5%
sparse	99	48.5%
full disclosure	51	25.0%

**Table 4 healthcare-10-01123-t004:** Hospitalisation due to COVID-19.

Hospitalisation due to COVID-19	Number	Percentage
not	187	91.7%
yes	17	8.3%

**Table 5 healthcare-10-01123-t005:** Level of depression in the studied population.

Depression	$\bar{x}$	Me	*s*	Min	Max
Beck Depression Rating Scale (points)	7.2	3	9.2	0	38

**Table 6 healthcare-10-01123-t006:** Classification of depression level according to norms.

Depression Level	Number	Percentage
no	139	74.3%
moderate	43	23.0%
severe	5	2.7%

**Table 7 healthcare-10-01123-t007:** Course of infection and depression.

Depression	Mode of Transmission of Infection						*p*
	Asymptomatic or No Infection (*n* = 54)		Oligosymptomatic (*n* = 99)		Fully Symptomatic (*n* = 51)		
	Mean	Median	Mean	Median	Mean	Median	
**Beck Depression Rating Scale (points)**	1.3	0	6.3	2.5	15.6	16	0.0000

*p*-test probability value calculated using Kruskal–Wallis test.

**Table 8 healthcare-10-01123-t008:** Need for hospitalisation and depression.

Depression	Hospitalisation due to COVID-19				*p*
	No (*n* = 187)		Yes (*n* = 17)		
	Mean	Median	Mean	Median	
**Beck Depression Rating Scale (points)**	6.2	2	17.9	21	0.0000

*p*-test probability values calculated using Mann–Whitney test.

**Table 9 healthcare-10-01123-t009:** Gender and level of depression.

Depression	Gender				*p*
	Woman		Man		
	Average	Median	Average	Median	
**Beck Depression Rating Scale (points)**	9.4	7.5	4.3	0	0.0000

*p*-test probability values calculated using Mann–Whitney test.

**Table 10 healthcare-10-01123-t010:** Age and level of depression—Spearman’s rank correlation coefficients (along with an assessment of the statistical significance).

Depression	Sex	
	Woman	Man
	Age	
**Beck Depression Rating Scale (points)**	0.07 (*p* = 0.4443)	0.45 (*p* = 0.0000)

## Data Availability

The data presented in this study are available on request from the authors.

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
