# Peer review of "Level of Depression during the COVID-19 Pandemic in Poland—A Cross-Sectional Study"

_healthcare, 2022, doi:10.3390/healthcare10061123_

Round 1

Reviewer 1 Report

This study aims to examine the depression level during COVID-19 pandemic in Poland. A cross-sectional study was conducted. I have several concerns about this study that the authors need to address in their revision:

  1. First, the literature review was limited and unclear. The rationale and motivation of this study is weak. There is a need for the authors to present research gaps in the literature and highlight how the current study can contribute to the literature.
  2. The aim of the study was vague and weakly related to the design and method of the study. Due to the cross-sectional design of this study, the study was not suitable to examine level of depression during COVID-19 pandemic as there is no reference point to compare. I would suggest the authors to reframe the introduction to focus more on demographic factors that are associated with depression during COVID-10 pandemic in Poland, rather the absolute levels of depression.
  3. In the introduction, the authors wrote: "A review of the literature shows that there are a number of causes that may influence the onset of anxiety or even depression [12,13,14,15]". This should be elaborated further.   
  4. The sample size is too small for a study that aimed to examine the prevalence of depression in a country-level. This should be highlighted in the limitation
  5. The authors should also highlight potential sampling bias that can affect the interpretation of the results. There is a need to elaborate the inclusion and exclusion criteria of sample recruitment. There is also a need to explain further the representativeness of the sample.

Author Response

Responses to Reviewer 1.

Kind regards, 

Katarzyna Van Damme-Ostapowicz

Reviewer 2 Report

This study examined the burden of depression and risk factors among a Polish sample during the COVID-19 pandemic. The assessment was conducted separately for men and women. The findings provided some research and clinical implications for mental illness prevention and treatment in Poland. 

However, there are several issues that should be addressed. 

Introduction:

1. The authors provided a good review of the general mental health burdens during the COVID-19 pandemic. However, there has not been sufficient discussion the specific hypotheses that this paper tests. What has been the level of depression in the existing body of literature? What are some of the risk factors for depression in the general Polish population? Any gender differences? A brief review of previous literature on these issues will provide some context for this study.

2. The authors should also briefly outline the specific research question and study hypotheses of this study at the end of the intro section.

Methods

1. Measurement of COVID-19 diagnosis: did the questionnaire ask if the participants were officially diagnosed with COVID-19? While I understand the use of symptoms to categorize the status of participants, an official diagnose (or test results) would provide useful information. If the authors did collect that information, how was it used in the analysis?

3. I found the categorization of COVID-19 status based on the number of symptoms reported slightly problematic. What were the rationale and justification using this method? Was this done in other studies? What did the authors mean when they said that they found this to be the most effective? Please elaborate.

4. The use of the online data collection methods was justified given the situation. However, was the sample representative of the general population in Poland? If no (which is very likely), how was the sample biased in terms of gender, age, socioeconomic status? This should be addressed in the discussion section. 

5. Table 10: the authors should specify what statistics were being reported in this table. What coefficients. A table should be able to stand alone, with enough information provided in the table title or footnotes.

6. In Line 94, the authors mentioned physical activity. I did not see any analysis on physical activity. Please explain.

7. Did the survey collect any socioeconomic status information and modifiable lifestyle behaviors such as alcohol use and tobacco use? Along with physical activities, these factors have been widely found to be related to depression.

A major issue is the sampling methods. How were the survey distributed? Was there a mailing list that the researchers used to distribute the survey via email? This information is crucial for the readers to decide the profile of the target population and the quality of the sample. A lot more information is needed. 

Author Response

Responses to Reviewer 2.

Kind regards, 

Katarzyna Van Damme-Ostapowicz

Reviewer 3 Report

Thank you for an opportunity to comment on a cross-sectional study on level of depression during the COVID-19 pandemic in Poland. It's a relatively simple cross-sectional study (as mentioned by authors in the limitations section), which provides important information on mental health (depression) in the Polish population during the Covid-19 pandemic. 

Several comments and suggestions for the authors:

  1. Abstract, line 16-17: please clarify what is meant by had "the best results" and "the worst results".
  2. Introduction, lines 38-41: please, provide references for statements "It can contribute to changes in the human psyche, including the emergence of generalised fear, increased anxiety or even the occurrence of depressive syndromes requiring the use of professional psychological care."
  3. Introduction, line 46 and 49: please, provide a clear rationale for the study, especially the choice of depression as a mental health variable of interest.
  4. Participants and Procedure, line 59-61: please, provide a reference for the choice of Covid-19 symptoms as a diagnostic tool for Covid in the study. Line 62: pleas, provide a rationale for the decision to "From the questionnaires received, every fifth person was randomly selected". Also, what method of randomisation was used in the study? Line 64-66: please, provide information (and references) how representativeness of the study sample was ensured ("The studied population is a representative sample for the population of recovered patients in Poland within age range from less than 25 to more than 65").
  5. Line 79: please, provide reference for "Beck Depression Inventory (BDI), was created by Aaron T. Beck."
  6. Statistical methods, line 91: please, explain what is meant by "differences among groups" - what groups? Line 95: provide information on "measures of physical activity".
  7. Results: please provide information on how data on "Mode of infection transmission", "Hospitalization due to COVID-19", and "Course of infection" were collected. Please, change phrase "psychological/mental condition/well-being" in the Results section to "depression". 
  8. Discussion, line 150-166: the first two paragraphs of Discussion are more suited for Introduction as they present the prevalence of depression during the pandemic and provide the rationale for the current study.  
  9. Discussion, line 185-186 and following: please, provide a possible explanation for the study results on gender, age and level of depression in the Polish population. It seems that results of the current study are different from outcomes of previous research, which is intriguing. 

Author Response

Responses to Reviewer 3.

Kind regards, 

Katarzyna Van Damme-Ostapowicz

Round 2

Reviewer 1 Report

The revision was acceptable. However, I am still not convinced that the sample is representative. I believe that there is a need for the authors to clarify further regarding their sample recruitment. 

Author Response

Dear Sirs, 

Thank You very much for your valuable comments on the article. We are very grateful to you for taking the time to assess our manuscript and for their constructive comments.

We hope that applied the amendments will enable the publication of the manuscript in your Journal.

We are looking forward to hearing from you.

Yours faithfully, 

Katarzyna Van Damme-Ostapowicz
